# Tissue Niches Formed by Intestinal Mesenchymal Stromal Cells in Mucosal Homeostasis and Immunity

**DOI:** 10.3390/ijms23095181

**Published:** 2022-05-06

**Authors:** Maria Pasztoi, Caspar Ohnmacht

**Affiliations:** Institute of Allergy Research, Centre of Allergy and Environment, Technical University and Helmholtz Centre Munich, Ingolstaedter Landstrasse 1, 85764 Neuherberg, Germany

**Keywords:** mesenchymal cells, fibroblasts, intestinal lamina propria, mucosal immunity, tissue niches, tolerance, stromal cells

## Abstract

The gastrointestinal tract is the largest mucosal surface in our body and accommodates the majority of the total lymphocyte population. Being continuously exposed to both harmless antigens and potentially threatening pathogens, the intestinal mucosa requires the integration of multiple signals for balancing immune responses. This integration is certainly supported by tissue-resident intestinal mesenchymal cells (IMCs), yet the molecular mechanisms whereby IMCs contribute to these events remain largely undefined. Recent studies using single-cell profiling technologies indicated a previously unappreciated heterogeneity of IMCs and provided further knowledge which will help to understand dynamic interactions between IMCs and hematopoietic cells of the intestinal mucosa. In this review, we focus on recent findings on the immunological functions of IMCs: On one hand, we discuss the steady-state interactions of IMCs with epithelial cells and hematopoietic cells. On the other hand, we summarize our current knowledge about the contribution of IMCs to the development of intestinal inflammatory conditions, such as infections, inflammatory bowel disease, and fibrosis. By providing a comprehensive list of cytokines and chemokines produced by IMCs under homeostatic and inflammatory conditions, we highlight the significant immunomodulatory and tissue niche forming capacities of IMCs.

## 1. Introduction

Tight regulation of intestinal immune responses is key in maintaining tolerance against harmless food and commensal-derived antigens while supporting effective immunity towards invading pathogens. A vast array of cellular and molecular mechanisms evolved to contribute to this delicate balance [1,2]; however, the precise mechanisms of intestinal immunity and homeostasis are still incompletely understood. Despite our expanding knowledge about epithelial and hematopoietic cells, there is still much to be revealed regarding the significance of non-hematopoietic mesenchymal cells in mucosal immunity of the gastrointestinal tract.

The term “mesenchyme” is assigned to multipotent mesodermal cells that give rise to various connective tissues during development. In adulthood, mesenchymal cells (MC) represent the non-hematopoietic, non-epithelial, and non-endothelial stromal compartments of connective tissues. They include highly diverse populations of fibroblasts, myofibroblasts, pericytes, smooth muscle cells, and mesenchymal progenitors [3,4]. In the intestinal tract, MCs are diffusely located in sub-epithelial compartments, such as the *lamina*
*propria* (LP) located directly beneath the epithelial barrier, the deeper *muscularis*
*mucosae*, the *submucosa*, and the most external intestinal layer called *serosa* [5]. Although intestinal MCs (IMCs) are primarily responsible for forming a structural framework in the gut wall, an increasing body of evidence suggests they harbor significant immunomodulatory functions [4,6,7]. Beside their interactions with epithelial and intestinal stem cells (ISC), the dynamic crosstalk between IMCs and hematopoietic cells attracted great attention recently [8,9]. Therefore, in this review, we will focus on the recently described cellular and immunological functions of IMCs and delineate their role in the maintenance of intestinal homeostasis and the development of intestinal inflammatory conditions. Although we will highlight recent findings on newly identified IMC subsets when available, the main focus of this review will be on the most extensively studied IMC populations of fibroblasts and myofibroblasts.

## 2. Heterogeneity of Intestinal Mesenchymal Cells

For a long time, the intestinal LP and the submucosa were considered to be predominantly made up of fibroblasts defined by the lack of the hematopoietic marker CD45 and α-smooth muscle actin (αSMA) expressing myofibroblasts [3,4,6]. However, recent single-cell RNA sequencing (scRNA-Seq) and high-resolution microscopy studies made great steps toward a better understanding of the heterogeneity of IMCs and revealed phenotypically distinct stromal populations occupying defined intestinal niches (Figure 1A) [5].

Directly under the epithelial barrier, subepithelial myofibroblasts (SEMFs), also called telocytes possess hundred-micrometer-long cytoplasmic extensions named `telopodes` and form an adjacent and contractile envelope from the crypt bottom to the tip of the villi [10,11,12]. The distribution of telocytes along this crypt–villus sheath is unequal, showing higher densities at the villus base and the tips [12]. Although no exclusive molecular markers have been identified so far, SEMFs or telocytes can be defined as platelet-derived growth factor receptor alpha (*Pdgfra^hi^*) [13] smooth muscle actin alpha 2 (*Acta2^lo^*) and forkhead box L1 (*Foxl1^+^*) cells [5] also expressing GLI family zinc finger 1 (*Gli1)* [14] and chondroitin sulfate proteoglycan 4 (*Cspg4)* (Table 1) [15]. Moving radially inward, one encounters CD31^+^Lyve1^−^ vascular endothelial cells (Va-EC) associated with *Cspg4^+^Pdgfrb^+^* pericytes (Table 1). In deeper layers of the intestinal LP desmin (*Des^+^*) myosin heavy chain 11 (*Myh11^+^*) *Acta2^hi^* myocytes are organized within tree-like smooth muscle fibers on vascular and lymphatic vessels throughout the entire villus length, finally merging with *submucosal* muscle cells (Table 1) [5,16]. Pdgfra^lo^ stromal cells or “interstitial fibroblasts” represent another abundant and still heterogeneous population of the intestinal LP characterized by the expression of CD34, podoplanin (PDPN, also called gp-38), and GLI1 (Table 1) [10,12,17]. Among Pdgfra^lo^ “interstitial fibroblasts”, gremlin 1 (*Grem1^+^*) atypical chemokine receptor 4 (*Ackr4^+^*) *CD81^+^* trophocytes form a distinct stromal population which can be also distinguished based on their localization between the external muscle and the crypt bottom (Table 1) [12,18,19]. Finally, CD31^+^Lyve1^+^ lymphatic ECs are found furthest from the epithelium (Table 1). 

According to the current literature, stromal populations identified in the colonic and small intestinal mesenchyme show substantial overlaps [14,20,21]. However, the specificity of certain markers slightly differs between colonic and small intestinal stromal subsets. For example, CD34 is not expressed by human colonic Pdgfra^lo^ cells, while CD81 represents a much broader marker for human colonic Pdgfra^+^ stromal cells [20,21]. Overall, none of the above-mentioned markers are fully unique to a single stromal cell population and some of these markers, such as vimentin (*Vim*) and *Gli1* are widely expressed across stromal subsets (Table 1). Moreover, the functional heterogeneity of these newly defined intestinal stromal populations still needs to be explored.

## 3. Intestinal Mesenchymal Cells in Homeostasis

The fundamental role of IMCs includes the maintenance of intestinal architecture through the production of extracellular matrix (ECM) molecules, such as type I, II, and V collagens, fibronectin, and matrix remodeling enzymes including matrix metalloproteinases (MMPs) (Figure 1C) [22]. Beyond matrix deposition, IMCs secrete diverse microenvironmental factors defining tissue niches required for the maintenance of epithelial and immune cell homeostasis (Figure 1B).

### 3.1. The Intestinal Mesenchymal–Epithelial Cell Axis in Homeostasis

The intestinal tract is covered by a single layer of epithelial cells. These cells originate from Lgr5^+^ ISCs found in the crypt base and continuously migrate towards the villi renewing crypts every 4–5 days [23,24]. During their migration, epithelial cells differentiate into specialized enterocytes, enteroendocrine cells, tuft cells, goblet cells, M cells, or Paneth cells [25]. All these specialized epithelial cells are not only responsible for nutrient absorption, but also form a tight barrier and represent the first line of defense against pathogens; thus making the maintenance of epithelial homeostasis particularly important. Since IMCs are localized in the sub-epithelial compartment, they closely interact with epithelial cells and sustain their proliferation and homeostatic functions [26]. This support largely relies on the secretion of mesenchyme-derived soluble factors of the Wnt and the bone morphogenic protein (BMP) family [27,28]. While canonical Wnt and R-spondin (RSPO) signaling promotes [29], BMPs attenuate the proliferation of ISCs [30,31]. Recent scRNA-Seq studies identified SEMFs/telocytes and subcryptal trophocytes as major cellular sources of these factors generating gradients of these signaling molecules from the top to the bottom of the crypts (Figure 1B) [5,12]. SEMFs/telocytes predominate in the upper crypt regions representing an abundant source of BMPs and the only source of BMP7, thereby creating a zone of high BMP activity [12,32,33]. In contrast, trophocytes localized in the basal part of the crypt express BMP inhibitors (BMPi), such as high levels of *Grem1*, the canonical *Wnt2b*, and all *Rspo*s (*Rspo1^lo^, Rspo2^lo^,* and *Rspo3^hi^*) [12,34]. These CD81^+^Pdgfra^lo^ stromal cells well deserve their name “trophocyte”, as they can efficiently replenish key trophic factors during in vitro organoid cultures due to their high expression of *Grem1* and *Rspo3* [12]. Beside these two stromal cell subsets, low expression of *Bmp* transcripts can be found in Pdgfra^lo^ stromal cells serving as another rich source of BMPi, canonical *Wnt2b,* and *Rspo3* [5,12]. Taken together, WNT2B, RSPO, and BMPi expressing subepithelial IMCs represent the minimal ISC niche with Wnt and BMPi activity dominating in the crypt bottom, while BMP activity is characteristic for the upper crypt regions (Figure 1B). 

Another recent study described the mouse colonic ISC niche supporting MC population as Gli1^+^CD90^+^ fibroblasts located around the bottom of the crypt. Apart from the secretion of Wnt ligands and BMPi, these fibroblasts produce class 3 semaphorins (Sema3) whereby promoting the proliferation of ISCs through binding to the epithelial neuropilin 2 receptor (Nrp2) [35]. Whether the proliferation-inducing property of Gli1^+^CD90^+^Sema3^+^ fibroblasts is achieved by the ability of Sema3a to increase Wnt activity [36] could not be proved in this study, probably due to the low number of Lgr5^+^ cells in the organoid cultures. 

Others reported that the intestinal ISC niche is supported by Lepr^+^ MCs located around the intestinal crypts via the stromal IGF1–epithelial IGF1R axis. Interestingly, the capacity of Lepr^+^ MCs in modulating ISC function during intestinal homeostasis and regeneration could be modulated by diet-related alterations [37]. Therefore, these findings provide the first link between diet-induced changes and IMC-driven functions.

In the human colon, a CD142 expressing subepithelial fibroblast subset was shown to be enriched for numerous BMP/Wnt ligands and was found in the proximity of epithelial crypts, thereby representing the key human stromal population for the maintenance of epithelial homeostasis [20].

Noteworthy, the interaction between mesenchymal and epithelial cells is bi-directional. While IMCs support epithelial differentiation and regeneration, epithelial-derived Indian hedgehog (Ihh) in turn regulates stromal development and proliferation [38]. In a more recent study, a defined subset of distal subepithelial Pdgfra^hi^ MCs was shown to activate Wnt signaling and stimulate the expression of Sonic Hedgehog (SHH) in epithelial cells, which in turn drive villi formation of the developing intestinal tract [13]. Moreover, epithelial-derived Ihh can also modulate IMC-driven intestinal immune responses by controlling the expression of stromal CXC motif chemokine ligand 12 (CXCL12) and the consequent migration of immune cells [39]. Collectively, the homeostatic crosstalk between mesenchymal and epithelial cells does not only impact barrier integrity but can additionally affect intestinal immune responses.

### 3.2. Intestinal Mesenchymal Cells Regulate Immune Cell Homeostasis

In addition to the interaction of stromal cells with epithelial cells, IMCs can directly crosstalk to immune cells under steady-state conditions through the secretion of different soluble mediators (Figure 1C) [7,40,41]. Among these factors, the release of different chemokines is of particular importance to establish a gradient for the migration and recruitment of a wide variety of leukocytes typically found within the LP, such as monocytes/macrophages, dendritic cells, granulocytes, NK cells, T- and B-lymphocytes (Table 2).

Interestingly, ECM molecules secreted by IMCs create a scaffold enabling the adherence and crawling of extravasating immune cells, while their glycosaminoglycan moieties bind extracellular chemokines whereby an immobilized chemokine gradient is established to support leukocyte migration [54]. Beyond chemokines, the expression of numerous immunomodulatory cytokines has been reported in IMCs (Table 3). Since most of our knowledge originates from scRNA-Seq expression studies or experiments with in vitro cultured stromal cells, the precise mechanisms of how and under which conditions different IMC-derived homeostatic cytokines and chemokines act on immune cells remain to be explored.

When considering key cellular interactions of IMCs under steady-state conditions, the crosstalk between stromal cells and T cells has gained high attention. As under steady-state conditions, the intestinal tract is a highly tolerogenic organ, intestinal stromal cells may be expected to play a critical role in the establishment and maintenance of this tolerogenic tissue microenvironment, similar to what was shown in the gut-draining mesenteric lymph nodes [66,67]. Namely, CD90^+^ colonic myofibroblasts promote in vitro conversion of naïve T cells into Foxp3^+^ regulatory T cells (Treg) in a major histocompatibility complex class II (MHC-II)- and prostaglandin E2-dependent manner [68]. Moreover, the expression of programmed death-ligands PD-L1 and PD-L2 molecules on steady-state human colonic CD90^+^ myofibroblasts and fibroblasts suppress the proliferation of activated CD4^+^ effector T cells, which might play a critical role in the maintenance of a tolerogenic tissue niche (Figure 1D) [69,70,71]. Another way to induce local tolerance can be accomplished by the capacity of gp-38^+^CD31^−^ intestinal LP stromal cells to induce the production of retinoic acid (RA) by CD103^+^ mucosal dendritic cells (Figure 1E) [56]. RA is generated during vitamin A metabolism by retinaldehyde dehydrogenase (RALDH) enzyme and plays a key balancing role in supporting Tregs while opposing Th17 differentiation in the intestinal mucosa [72,73]. While RA production and *Aldh1a2/3* expression by defined stromal subsets of the gut-draining mesenteric lymph nodes are known components of tolerance induction [1,66,74], the direct contribution of IMC-derived RA to mucosal tolerance remains largely unclear. Interestingly, the generation of tolerogenic CD103^+^ mucosal dendritic cells depends on intestinal LP stromal cell-derived RA and granulocyte-macrophage colony-stimulating factor (GM-CSF) [56], a major differentiation factor for multiple myeloid cell types, including granulocyte and monocyte/macrophage lineages. Intriguingly, mucosal dendritic cells, in turn, promoted GM-CSF production by LP stromal cells indicating the existence of bi-directional crosstalk [56].

Besides the interaction of IMCs with T cells and dendritic cells, in vitro studies revealed that gut LP stromal cells might also contribute to the formation of immunoglobulin A (IgA^+^) plasma cells by the release of B-cell activating factor (BAFF) (Figure 1E) [55,75]. However, the addition of anti-BAFF antibodies to the co-cultures did not fully block the formation of IgA^+^ plasma cells, indicating a role for other yet undefined stromal cell-derived soluble factors in regulating the stromal–B-cell axis in the steady-state intestinal mucosa.

Although it is far from being well understood, there are also indications for the crosstalk between IMCs and mast cells. In vitro co-culture of colonic IMCs with bone marrow-derived mast cells induced the expression of *Mcpt1* and *Mcpt2* genes, which are indicators of mucosal-type mast cells. In addition, the co-culture of bone marrow-derived mast cells with colonic IMCs led to the upregulation of the extracellular ATP purinoreceptor P2X7, a marker of mast cell activation [76]. Since inhibitors of RA receptor efficiently suppressed the IMC-driven upregulation of P2X7 in mast cells [76], there is a strong indication that colonic IMC-derived RA may control mucosal mast cell development.

In conclusion, the above studies reflect diverse crosstalk between IMCs and different immune cells whereby IMCs can contribute to the maintenance of intestinal homeostasis.

## 4. Intestinal Mesenchymal–Immune Cell Interactions upon Inflammatory Conditions

How the above-mentioned steady-state cellular interactions of IMCs contribute to the development of intestinal inflammatory disorders still needs to be understood in detail. However, there is now mounting evidence that IMCs, similar to MCs in other tissues, get stably and persistently activated upon exposure to inflammatory conditions [8,9,41]. This includes significant changes in their phenotype and functions, such as an altered capacity for tissue destruction, ECM production, responsiveness to inflammatory cytokines, interactions with immune cells, or the initiation of angiogenesis [6,7,77]. Therefore, in the next part of this review, we will discuss the current knowledge about the responsiveness of IMCs to inflammatory stimuli and the cellular and molecular alterations of IMCs during inflammatory bowel diseases (IBD), infections, tissue repair, and fibrosis (Figure 2). The role of cancer-associated fibroblasts (CAF) in gastrointestinal tumorigenesis represents another area of intense research reviewed in detail elsewhere [8,41], thus being outside the scope of this review.

### 4.1. Responsiveness of Intestinal Mesenchymal Cells to Microbial Stimuli and Inflammatory Cytokines

An increasing body of knowledge suggests that IMCs can respond to microbial stimuli due to the expression of pattern recognition receptors (PRR), such as toll-like receptors TLR1-9 and nucleotide-binding oligomerization domain-containing receptors NOD1-2 [4,7,58]. Most of the available studies, however, only investigate the responsiveness of IMCs to PRR ligands using in vitro cultured stromal cells, and very few publications provide direct in vivo evidence on the active contribution of IMCs to anti-microbial immune responses.

The impact of the TLR4 ligand lipopolysaccharide (LPS) on IMCs has been most widely investigated. Numerous studies showed that LPS stimulates the expression of interleukin 6 (IL-6), IL-8, chemokine ligand 2 (CCL2), CXCL1, and cyclooxygenase-2 (COX-2) in cultures of human and murine intestinal or colonic fibroblasts and myofibroblasts [57,59,61,78,79]. More recently, transcriptional analysis with LPS stimulated in vitro cultured intestinal stromal cells revealed the upregulation of further pro-inflammatory mediator-encoding genes, such as *IL1b*, *Tnf, Il12b*, *Cxcl9*, *Cxcl10*, and *Cxcl11* [80]. 

Other ligands activating TLR1, TLR2, and TLR6, such as Pam2CSK4, Pam3CSK4, and FSL-1 synthetic lipoproteins or heat-killed *Listeria*
*monocytogenes* similarly stimulated IL-6 and IL-8 secretion of intestinal fibroblasts [59,61]. In addition, in vitro cultured intestinal or colonic fibroblasts are also able to respond to the TLR5 ligand flagellin by increased secretion of IL-6 and IL-8 [59], IL-1α, IL-1β, IL-18, IL-33, and GM-CSF [58], upregulation of tumor necrosis factor (*Tnf)* expression [65], and increased fibronectin (FN) and type 1 collagen (COL1) production in a MyD88-dependent manner [81]. Thus, the activation of IMCs in response to microbial stimuli does not only lead to the secretion of diverse immunomodulatory mediators but also affects their capacity to produce diverse ECM components. 

As a piece of more direct in vivo evidence on the PRR-driven interaction between IMCs and intestinal pathogens, TLR2 expression on tissue-resident cells was found to mediate protection against *Citrobacter*
*rodentium* (*C.*
*rodentium*) induced lethal colitis [82]. This TLR2-dependent mucosal protection relied partly on the activation of STAT3 in epithelial cells (Figure 2A) [82], which is known to drive mucosal tissue repair [83]. Moreover, induced expression of the pro-fibrogenic cytokine IL-11 in αSMA^+^ myofibroblasts of the *muscularis*
*mucosae* was also part of the TLR2-dependent protection from mucosal damage upon *C.*
*rodentium*-induced colitis (Figure 2A) [82]. Others reported that colonic stromal cells can also contribute to the efficient eradication of *C.*
*rodentium* in a Nod2-dependent manner by increased CCL2 secretion and consecutive recruitment of inflammatory monocytes (Figure 2A) [43]. 

Additional recent evidence suggests that IMCs significantly contribute to intestinal immune responses upon systemic infection with *Salmonella*
*typhimurium (S. typhimurium)* [63]. In this study, the role of pericryptal fibroblasts-derived alarmin IL-33 was investigated, as this cytokine was shown to protect against the intestinal pathogen. Mechanistically, fibroblast-derived IL-33 induced essential changes in the epithelial barrier including the differentiation of stem and progenitor cells into secretory-type epithelial cells in a Notch-dependent manner and consequently promoting antimicrobial defense (Figure 2A) [63]. Altogether, current findings indicate that IMCs can sense and respond to microbial antigens; thus actively contributing to host defense mechanisms against intestinal pathogens.

Beside PRRs, IMCs express different cytokine receptors allowing them to respond to and integrate cytokine signals through the release of further inflammatory cytokines and chemokines (Table 2 and Table 3). In vitro studies revealed that IL-1α, IL-1β, IL-17, and TNF-α cytokines can induce the secretion of IL-6, IL-8, IL-33 cytokines, and CCL2 chemokine in colonic and intestinal myofibroblasts [43,57,60,61,62,64]. In addition, the activation of IMCs by different pro-inflammatory cytokines, such as IL-1β, IL-17, and TNF-α can also drive the secretion of ECM molecules, such as collagen types 1 and 4 [62,84], and the production of metalloproteinases including MMP1, MMP3, and TIMP1 [62,85]. An early report indicated a possible crosstalk between IMCs and myeloid cells upon inflammation by showing elevated M-CSF and GM-CSF secretion in TNF-α-stimulated colonic myofibroblasts [57]. Interestingly, stimulation of IMCs by IL-1β can have profound effects on their fibrogenic potential. When IL-1β was added together with TNF-α, the gene expression of *IL36γ*, a key cytokine in IBD-related fibrosis [53], was induced in human colonic myofibroblasts [86]. On the other hand, the combination of IL-1β and TGF-β synergistically enhanced the expression of the pro-fibrogenic cytokine IL-11 on intestinal SEMFs [87]. The expression of IL-11, among other inflammatory mediators and matrix remodeling enzymes, could also be stimulated in activated myofibroblasts by the IL-10 family member IL-22, another important player in the pathogenesis of IBD [88]. Importantly, IFN-γ was shown to be a key driver of MHC-II expression in human intestinal myofibroblasts [59], indicating the key role of IMCs in acting as non-professional antigen-presenting cells which has been reported both under steady-state [89] and inflammatory conditions [90].

In conclusion, though most of these stimulation experiments provide only an indirect link, they clearly indicate that IMCs play a critical role in the regulation of immune responses upon intestinal inflammation by responding to and producing a myriad of pro-inflammatory cytokines, immune cell attracting chemokines, and matrix remodeling molecules.

### 4.2. Intestinal Mesenchymal Cells in the Pathogenesis of Inflammatory Bowel Disease

The crosstalk of IMCs with immune cells under inflammatory conditions is increasingly appreciated but very little is known about the precise mechanisms of how activated IMCs contribute to the immunopathology of IBD (Figure 2B). TNF-α is the most extensively studied cytokine in the immunopathogenesis of IBD; consequently, anti-TNF-α therapies have been successfully introduced and are widely used in the clinics [77]. The activation of IMCs by epithelial cell-derived TNF-α is a necessary and sufficient step for disease development in the TNF^ΔARE^ mouse model of Crohn’s disease (CD) [91,92,93]. This stromal activation takes place well before inflammatory cell infiltration and results in the upregulation of intercellular adhesion molecule 1 (ICAM-1), MMP3, MMP9, and MMP13 on ileal CD90^+^αSMA^+^CD31^−^ cells [94]. Beside the key role of TNF-α, other pro-inflammatory cytokines are also involved in disease pathogenesis. For example, in a colitis model induced by the transfer of CD4^+^CD45RB^high^ cells into severe combined immunodeficient (SCID) mice, a subpopulation of colonic CD40^+^CD90^+^αSMA^−^ MCs was shown to secrete higher amounts of IL-6, CCL2, and CCL5 in response to interferon-γ (IFN-γ) and CD40L [42]. In addition, IL-17, highly expressed in CD patients, can directly stimulate intestinal fibroblasts by inducing gene expression of the transcription factor *NFKBIZ,* and the neutrophil-attracting chemokines *CXCL1* and *CXCL6* [52]. Taken together, these data highlight multiple cytokine-driven mechanisms of IMC activation, which in turn create an inflammatory gut microenvironment during IBD development via the secretion of further cytokines and immune cell attracting chemokines. 

Recent scRNA-Seq studies greatly contributed to a better understanding of how different IMC subsets and IMC-derived factors shape the inflamed intestinal microenvironment during IBD. First, Kinchen and colleagues described four major fibroblast-like subsets (S1–S4) and an additional population of myofibroblasts as part of the colonic stromal compartment in healthy controls and patients with ulcerative colitis (UC) [20]. The S1 cluster was defined by enrichment for non-fibrillar collagen and elastic fiber-encoding genes. Stromal cells in the S2 cluster expressed high levels of collagens forming the epithelial basal membrane and members of the *BMP* and *WNT* families; thus representing a key subset supporting epithelial stem cell proliferation and differentiation (main S2 markers: *CD142, POSTN*). Cluster S3 was identified by elevated expression of *CD55* and *COX2*. Cluster S4 was a tiny population in the healthy gut but expanded dramatically in patients with UC and showed upregulation of fibroblastic reticular cell-like genes such as *PDPN, CD74, CCL19,* and *IL33*. These observations reflect the possible role of S4 stromal cells in generating tertiary lymphoid follicles upon inflammation. Moreover, *Lox* and *Loxl1* lysyl oxidase encoding genes showed elevated expression levels in S4 cells from DSS-treated mice and the blockade of these enzymes efficiently reduced the severity of DSS colitis [20]. In another study, eight different fibroblast clusters were described by scRNA-Seq analysis in the colon of UC patients and healthy controls and were mainly distinguished by the expression of genes belonging to the *BMP* and *WNT* families [21]. Among these clusters, an inflammation-associated fibroblast (IAF) population was also identified which dramatically expanded in inflamed tissues of UC patients. IAFs were enriched for genes such as *IL11, IL24,* and *IL13RA2* and multiple CAF markers (*FAP, TWIST1*, and *WNT2*). More interestingly, IAFs and inflammatory monocytes from anti-TNF-α non-responders were highly enriched for Oncostatin M receptor (*OSMR*) and *OSM*, respectively. Therefore, these data indicate the role of IAFs both in the pathology of colitis and fibrosis and their implication in anti-TNF-α resistance. To what extent IAFs correspond to the expanded S4 cluster described by Kinchen and colleagues, needs further clarification. As expression of BMP and Wnt family members by IMC subsets was found to be a major determinant to define IMC clusters in both studies and their expression is not evenly distributed along the crypt–villi axis (Figure 1B), it will be important to investigate whether the individual IMC clusters can be linked to specific anatomical locations and their associated function.

Moreover, the heterogeneity of ileal MCs was also investigated in human CD [45]. Beside endothelial cells, pericytes, and smooth muscle cells, one fibroblast cluster was defined in this study based on the expression of platelet-derived growth factor receptors (*PDGFRA* and *PDGFRB*). Among them, an activated fibroblast subset was distinguished and their close interaction with myeloid cells was hypothesized. Due to the expression of cytokine receptors on these activated fibroblasts, they can be easily activated by inflammatory macrophage-derived cytokines such as TNF-α, IL-1β, and OSM. As a response pattern, activated fibroblasts upregulated neutrophil-attracting *CXCL1, CXCL2, CXCL5, CXCL8,* pro-fibrotic *IL11*, and monocyte-recruiting *CCL2* and *CCL7*. The latter two chemokines can then further facilitate monocyte recruitment to the site of inflammation as a positive feedback loop. The appearance of an inflammatory fibroblast population during human CD was reinforced in further upcoming scRNA-Seq studies [44,50]. In the latter, the crosstalk between stromal and epithelial cells was highlighted via stromal-derived *WNT2* and the pro-apoptotic *TNFSF10*-*TNFRSF10B* axis [50]. These findings clearly indicate an important role of IMCs in the regulation of intestinal epithelial cell death and consequential epithelial barrier dysfunction which is a key characteristic of CD.

*NOD2* mutations were previously linked to CD as risk factors [95]. Recently, the presence of *NOD2* mutations in CD patients was described as affecting the crosstalk between fibroblasts and macrophages [46]. *NOD2* mutations imprinted an activated fibroblast signature characterized by increased expression of *IL11* and Wilms tumor 1 (*WT1*). Moreover, the expression of gp130 ligands, such as *IL6, IL11,* and *OSM* was even more increased in activated fibroblasts of anti-TNF-α non-responder patients [46]. *OSM* was previously described as a key signaling molecule in driving intestinal inflammation and involved in resistance towards anti-TNF-α therapy probably through signaling via OSMR of non-epithelial stromal cells [47]. Therefore, the blockade of gp130 family members represents an alternative treatment for these selected CD patients by breaking up the circuit of fibroblast-immune cell interactions. 

Beside the crosstalk to immune cells, the interaction between IMCs and epithelial cells represents another layer in the immunopathology of IBD. Especially, the importance of the Hedgehog (Hh) signaling pathway was more extensively studied in the communication between these two cell types. Genetic and pharmacological inhibition of the Hh pathway was shown to intensify the severity of DSS-induced colitis in mice partly due to reduced IL-10 expression in CD45^−^Gli1^+^ stromal cells and the decreased number of CD4^+^Foxp3^+^ T cells in the colonic LP [96]. Moreover, the lack of epithelial cell-derived Ihh led to increased secretion of fibroblast-derived CXCL12, followed by further recruitment of immune cells into the LP and increased sensitivity to DSS-induced colitis [39]. Thus, disruption of the stromal–epithelial crosstalk may lead to the loss of immunosuppressive IMC phenotype and rather imprint an inflammatory program in IMCs during IBD development. In line with these findings, human colonic CD90^+^ (myo)fibroblasts from IBD patients were reported to possess a reduced Treg inducing capacity [68]. Furthermore, human CD90^+^ (myo)fibroblasts from inflamed CD colon showed reduced expression of PD-L1 when compared to healthy controls or non-inflamed CD colon and harbored a lower capacity to suppress Th1 cell activity [70]. 

More recently, the concept of mesenchymal plasticity has been introduced in the context of stromal–epithelial crosstalk both during homeostasis and upon DSS-induced colitis [97]. Melissari and her colleagues described the role of Col6a1^+^CD34^−^Pdgfra^hi^ telocytes in the regulation of enteroendocrine cell differentiation and epithelial proliferation under steady-state conditions. Importantly, depletion of these cells by using Col6a1^DTR^ mice resulted in the activation and increased proliferation of Col6a1^−^CD34^+^ stromal cells both during homeostasis and upon DSS-induced colitis. Therefore, the occupation of the space of depleted telocytes at the top of the colonic crypts by Col6a1^−^CD34^+^ stromal cells provided evidence for the plasticity of IMCs.

Despite these advances, there are still many open questions about which particular signal(s) drive the loss of the immunosuppressive properties of IMCs during IBD development. Future studies should clarify whether tolerogenic properties can be confined to unique subsets of IMCs. Nevertheless, understanding the mechanism and key drivers of mesenchymal plasticity could pave the way for more specific IMC-targeting approaches in the treatment of IBD.

### 4.3. Intestinal Mesenchymal Cells in Tissue Repair and Fibrosis

Tissue repair, such as wound healing is inevitable and necessary for the termination of acute intestinal inflammatory responses. Through the production of ECM molecules, matrix remodeling enzymes, inflammatory mediators, and the activation of precursor cells, activated fibroblasts, and αSMA^+^ myofibroblasts support wound healing at the epithelial barrier. Once the inflammation is resolved and tissue repair was successful, the phenotype of activated cells returns to their homeostatic status. Upon chronic non-resolving inflammation, tissue repair becomes uncontrolled in a vicious circle and can result in fibrosis characterized by further activation of myofibroblasts, increased production and deposition of ECM molecules, and tissue damage (Figure 2C) [41,98,99]. 

The key driver of myofibroblast activation is TGF-β. This cytokine promotes the differentiation of fibroblasts into myofibroblasts, upregulates αSMA expression, promotes proliferation [100], supports MCs matrix remodeling capacity by inducing MMP-9 and collagen production [101], and can induce the expression of additional fibrogenic response genes, such as *COL1A1, FN1, ACTA2, MKL1*, and *MYLK* [102]. Beside TGF-β, several other mediators and factors were identified in the pathogenesis of intestinal fibrosis [98,103].

For example, the pro-fibrotic cytokine IL-11 was shown to be produced by myofibroblasts, smooth muscle cells, and damaged epithelial cells. Both smooth muscle and stromal IL-11 secretion were sufficient for driving fibrosis during IBD when using transgenic mice with conditional *Il11* overexpression [104]. Since *Il11* expression is highly upregulated in anti-TNF-α therapy of non-responders compared to responder IBD patients [105], it represents a promising therapeutic target. Thus, under which circumstances IMCs upregulate IL-11 and its mode of action during IBD-related fibrosis should be investigated more extensively.

Another potential mediator of fibroblast activation and fibrosis during IBD includes the IL-1 cytokine family member IL-36. Initially, elevated expression of IL-36 was reported in colonic tissue samples of IBD patients with fibrotic stenosis formation (fibrostenosis), which highly correlated with the presence and increased number of adjacent αSMA^+^ cells. Importantly, stimulation of human primary colonic fibroblasts via the IL36R resulted in their long-lasting pro-fibrotic activation [53]. 

More recently, the CSF-1R ligand IL-34 was shown to be an additional activator of intestinal fibroblasts during fibrosis. In vitro administration of IL-34 to fibroblasts from control tissues promoted wound healing and led to the upregulated expression and secretion of COL1A1 and COL3A1 [106], indicating an important role of IL-34-driven fibroblast activation upon intestinal fibrogenesis. However, cultured human intestinal fibroblasts of CD patients with fibrostenosis themselves showed an elevated IL-34 expression compared to fibroblasts isolated from control ileal tissues [106], suggesting that activated intestinal fibroblasts cannot only be stimulated by but also serve as a reservoir of IL-34. Whether intestinal fibroblast-derived IL-34 acts in an autocrine or paracrine manner during the pathogenesis of fibrostenoic CD awaits further clarification.

IL-33 is an important key cytokine/alarmin for the maintenance of epithelial barrier integrity and was recently reported to be produced by tissue-resident MCs of different origins [107,108,109] including steady-state fibroblasts of the intestinal LP [20,35]. In an experimental model of intestinal fibrosis driven by an adherent-invasive *E.Coli* (AIEC) strain, the blockade of IL-33-ST2 (also known as Il1rl1, the receptor for IL-33) signaling was sufficient to suppress fibrosis and decrease collagen deposition. However, the authors in this study could not prove the direct AIEC-driven induction of collagen in human primary colonic fibroblasts, nor could they identify fibroblasts as a major source of IL-33 [65]. More recently, αSMA^+^ myofibroblasts and vimentin^+^ fibroblasts were identified as primary sources of colonic IL-33 and both subsets were expanded upon DSS-induced colitis [64]. Overall, the precise contribution of IMC-derived IL-33 to intestinal fibrosis remains obscure.

Another novel player of intestinal fibrosis is TL1A encoded by the *TNFSF15* gene. TL1A was shown to bind to death domain receptor 3 (DR3) preferentially expressed on vimentin^+^αSMA^+^ primary intestinal myofibroblasts and induced the expression of *Col1a1* in these cells [110]. In vivo administration of TL1A neutralizing antibodies reversed colonic fibrosis partly by reducing the number of colonic fibroblasts and myofibroblasts [110], while transgenic TL1A overexpression led to the development of fibrostenosis [111]. In a follow-up study, the intestinal microbiota was shown to be required for TL1A-induced fibrosis and related fibroblast activation [112]. The link to microbiota was recently reinforced by others showing that αSMA-specific deletion of MyD88 reduces the severity of DSS colitis-associated intestinal fibrosis [81]. Still, further studies are required to better understand the mechanism whereby microbiota regulate fibroblast-driven fibrotic pathologies. 

Importantly, the inflamed mucosa of IBD patients shows elevated levels of many more pro-fibrogenic factors, such as PDGF, epidermal growth factor (EGF), basic fibroblast growth factor (bFGF), insulin-like growth factor-1 (IGF-I), and fibronectin [98]. Beyond their fibrogenic potential, these mediators can also stimulate in vitro myofibroblast migration [113]. In addition, several pro-inflammatory cytokines, such as IL-1, IL-6, IL-13, IL-17A, TNF-α, and IFN-γ were reported to be involved in the regulation of fibrogenic functions of fibroblasts and myofibroblasts [98]. 

Notably, epithelial and endothelial cells can directly give rise to invasive and migratory fibroblasts in vivo during tissue fibrosis. This TGF-β-driven process is called epithelial- or endothelial-to-mesenchymal transition (EMT) and represents the loss of typical epithelial/endothelial markers together with the upregulation of fibroblast markers, such as αSMA, vimentin, fibronectin, and collagens, and the acquisition of a fibroblast-like morphology [103,114]. Although the occurrence of EMT in IBD-related intestinal fibrosis was previously reported [115,116], our understanding of its functional role and the exact mechanism is still in its infancy.

## 5. Conclusions and Future Directions

Although unraveling the role of IMCs in the maintenance of mucosal homeostasis and intestinal inflammatory conditions has just started, current research is already showing their considerable contribution beyond pure scaffold and trophic function. In this review, we focused on the heterogeneity of IMCs under steady-state conditions and when available we discussed the role of distinct stromal subsets in disease development.

Still, numerous challenges make the investigation of intestinal stromal cells difficult at present. One of the major challenges is the lack of agreement on the definition of IMCs and all of the newly identified subsets. Using different markers to define IMC populations makes it difficult to form a comprehensive picture, as it remains uncertain whether different studies investigate the same IMC population. Additional variability may derive from the exact anatomical location and differences in isolation procedures. Furthermore, most of our knowledge originates from observations with in vitro cultured stromal cells without considering that long-term culture might drastically change their phenotype. Many of these studies did not provide information on the purity of in vitro cultured stromal cells, although only few contaminating cells can easily bias the result of gene expression analyses. According to our experiences, tissue-resident macrophages, for example, can easily contaminate short-term in vitro stromal cell cultures due to their long-lived nature. Nevertheless, the isolation protocol itself and the use of different digestion enzymes can also profoundly affect the composition of these in vitro cultures and the representation of different IMCs subsets.

Recent studies using single-cell technologies revealed a previously assumed heterogeneity of IMCs; however, it remained obscure whether these newly described IMC subsets represent functionally distinct populations. To unravel this critical question, extensive in vivo studies and the development of novel genetic and imaging tools are required. Especially, the generation of temporal- and lineage-specific depletion models could help to evaluate the role of these populations in disease development and homeostasis. For example, the recently described pro-inflammatory subset of IAFs was shown to expand upon IBD; however, their contribution to disease development or their cellular interactions during pathology needs to be further explored. In general, a molecular understanding of how IMCs and immune cells work in tandem to achieve homeostasis or drive intestinal inflammation might require the implementation of in vitro organoid co-culture models in the future. In addition, more studies are needed to explore whether and how the wide variety of IMC-expressed chemokines guide leukocyte trafficking under steady-state or inflammatory conditions.

Although IMCs were shown to express a diverse set of PRRs and actively respond to different TLR ligands, due to the lack of extensive in vivo studies, it remains unclear how stromal recognition of microbial antigens contributes to immune responses. It also needs to be investigated whether PRR-activated IMCs could possess an “innate-like inflammatory memory”, similar to what was described as trained immunity in the case of myeloid or epithelial cells [117,118]. More importantly, few studies have already suggested the impact of the microbiota or commensal-derived metabolites on certain IMC functions [56,81,112,119], but it remains unclear how IMCs have access to these TLR ligands, especially under homeostatic conditions, when the epithelial barrier is intact. Thus, a better understanding of the crosstalk between IMCs and commensal bacteria might provide unexpected insight into the development of intestinal inflammatory conditions. Altogether, novel technologies have led to the discovery of unprecedented heterogeneity among IMCs to unravel novel molecular insights into the diverse functions of IMCs. In consequence, this knowledge will hopefully allow in the future a more specific targeting of inter-cellular crosstalk between IMCs and hematopoietic or epithelial cells, especially under chronic inflammatory conditions such as IBD or fibrosis.

## Figures and Tables

**Figure 1 ijms-23-05181-f001:**
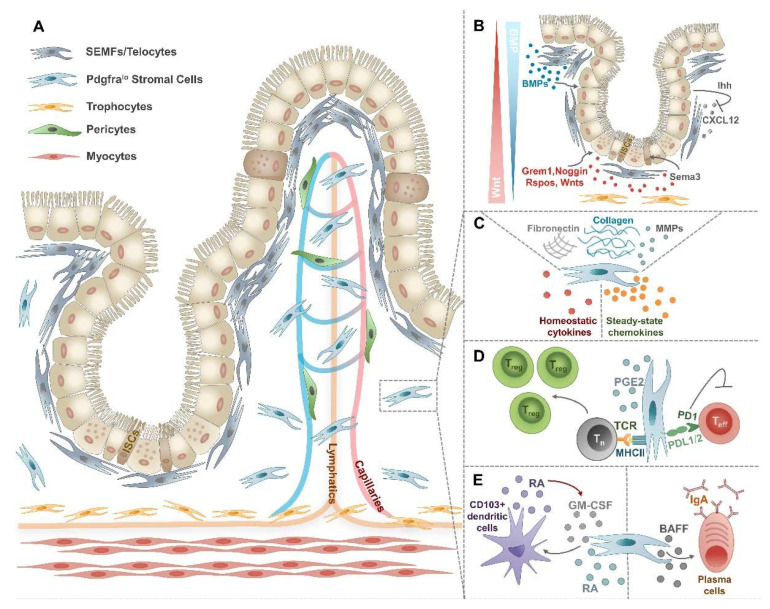
Heterogeneity and functions of steady-state IMCs. (**A**) In the *lamina*
*propria* and the *muscularis*
*mucosae* distinct subsets of IMCs can be found. Directly underneath the epithelium, SEMFs or telocytes form an envelope. Pdgfra^lo^ stromal cells are dispersed along the crypt–villus length. Close to the *muscularis*
*externa* and below the crypts, trophocytes form a stripe. Pericytes are found in the perivascular area. (**B**) SEMFs and trophocytes play a critical role in creating a niche for ISCs by the secretion of Wnt and BMPs. (**C**) Under steady-state conditions, IMCs not only excel in producing ECM components and matrix remodeling enzymes but also secrete a wide array of immunomodulatory cytokines and chemokines. (**D**) By the expression of PDL1/2 and MHC-II molecules, IMCs play a critical role in the maintenance of intestinal tolerance. (**E**) IMCs may induce tolerogenic CD103^+^ dendritic cells and IgA^+^ plasma cells, both of which contribute to intestinal homeostasis.

**Figure 2 ijms-23-05181-f002:**
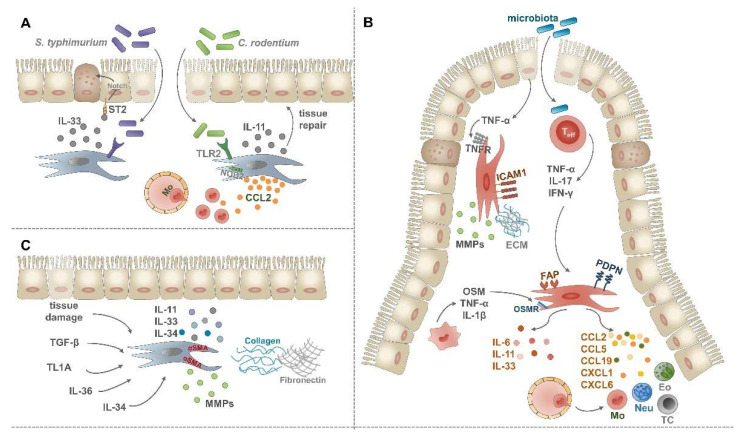
Role of IMCs upon intestinal inflammatory conditions. (**A**) Following infection with *C.*
*rodentium* or *S.*
*typhimurium* IMCs become activated. By the production of IL-11 and IL-33 pro-fibrogenic cytokines and CCL2 chemokine, they actively support tissue repair and the recruitment of inflammatory monocytes. (**B**) Fibrosis can be induced by dysregulated wound healing and a number of secreted factors, such as transforming growth factor-β (TGF-β), TL1A, IL-36, and IL-34. These mediators drive the accumulation and differentiation of αSMA^+^ myofibroblasts which then secrete excess ECM molecules and pro-fibrogenic cytokines. (**C**) Upon IBD, fibroblasts can be activated by TNF-α of epithelial origin, leading to their increased ICAM-1 expression and fibrogenic potential. Activated T cells can serve as another source of pro-inflammatory cytokines, such as TNF-α, IL-17, and IFN-γ, driving fibroblast activation. As a response, activated fibroblasts secrete pro-inflammatory and pro-fibrogenic cytokines, a vast array of chemokines, upregulate the expression of FAP and podoplanin and interact with macrophages via the OSM-OSMR axis.

**Table 1 ijms-23-05181-t001:** Populations of intestinal mesenchymal stromal and endothelial cells. Expression of molecular markers in defined cell types based on bulk and scRNA-Seq studies. The number of “+” reflects their expression levels, while grey background indicates the lack of expression. SEMF—subepithelial myofibroblasts or telocytes; Ly-EC—lymphatic endothelial cells; Va-EC—vascular endothelial cells.

	SEMF	Tropho-Cytes	Pdgfra Low Stroma	Myocytes	Pericytes	Ly-EC	Va-EC
*Vim*	++	++	++	++	++		
*Gli1*	+	+	+	+			
*Pdgfra*	++	+	+				
*Foxl1*	+			+			
*Cd81*	+	++	+				
*Grem1*		++	+				
*Cd34*		++	++				
*Pdpn*	+	+	+			+	
*Ackr4*		+		+			
*Des*				++	++		
*Acta2*	+			++	+		
*Myh11*	+			++	+		
*Pdgfrb*					++		
*Cspg4*	+		+		++		
*Lyve1*						++	
*Cd31*						++	++

**Table 2 ijms-23-05181-t002:** List of chemokines expressed by IMCs under steady-state and inflammatory conditions. Target cell information is based on the Uniprot database. Ba—basophils; BC—B cells; DC—dendritic cells; Eo—eosinophils; HSC—hematopoietic stem cells; Ly—lymphocytes; Mo—monocytes; Mac—macrophages; Neu—neutrophils; NK—NK cells; TC—T cells.

Chemokines	Target Cells	References— Steady-State	References— Inflamed
CCL2	Mo, Ba	[17,42,43,44]	[17,42,43,44,45,46,47]
CCL3	Mo, Mac, Neu	NA	[42]
CCL5	Mo, TC, Eo	[17,42,43,44]	[42,48]
CCL7	Mo, Eo	[21]	[21,45]
CCL8	Mo, Ly, Ba, Eo	[20,21,35,44,49]	[20,21,35,44,45,49]
CCL11	Eo	[20,35,44,50]	[20,21,35,44,45,50]
CCL13	Mo, Ly, Ba, Eo	[20,21]	[20,21,45]
CCL19	Ly (TC, BC)	[17,20,21,44,50]	[17,20,21,44,50]
CCL21	TC	[44,50,51]	[20,21,44,50,51]
CXCL1	Neu	[17,20,21,44,52,53]	[17,20,21,44,45,47,52,53]
CXCL2	Neu, Ba, Eo, HSCs	[17,20,35,50]	[17,20,35,45,47,50]
CXCL3	Neu	[20]	[20,45,47]
CXCL5	Neu	[15,44,53]	[15,44,45,47,53]
CXCL6	Neu	[52]	[21,45,47,52]
CXCL8	Neu, Ba, TC	NA	[45]
CXCL9	Mac, NK, NKT, TC	NA	[47]
CXCL10	Mo, Mac, TC, NK, DC	[51]	[47,50,51]
CXCL11	Mo, Neu, TC	NA	[47]
CXCL12	Mo, Ly	[20,39,44,50]	[20,21,39,44,45,50]
CXCL13	Mo, Neu, BC	[17,35,46,50]	[17,20,35,45,46,50]
CXCL14	Neu, DC	[35,44,51]	[35,44,45,51]

**Table 3 ijms-23-05181-t003:** List of cytokines expressed by IMCs under steady-state and inflammatory conditions.

Cytokines	References—Steady-State	References— Inflamed
BAFF	[55]	NA
GM-CSF	[56]	[45,57,58]
IL-1α	NA	[58]
IL-1β	NA	[44,47,58]
IL-6	[17,19,20,35,42,46,53,57,59,60]	[17,20,44,45,46,47,50,53,57,59,60,61]
IL-7	[35]	[17]
IL-8	[57,59,60]	[57,59,60,62]
IL-11	[46]	[20,21,44,45,46,47]
IL-16	[35]	NA
IL-18	NA	[58]
IL-20	NA	[21]
IL-24	NA	[21,44]
IL-32	[20,44]	[44]
IL-33	[20,35,63,64]	[20,58,63,64]
IL-34	NA	[45]
M-CSF	NA	[57]
OSM	NA	[46]
TNF-α	NA	[65]
TNFSF11	NA	[21]
TNFSF13B	[20,35]	[20]
TNFSF14	[20]	[20]

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
