# Peer review of "Tissue Niches Formed by Intestinal Mesenchymal Stromal Cells in Mucosal Homeostasis and Immunity"

_ijms, 2022, doi:10.3390/ijms23095181_

Round 1
Reviewer 1 Report
The presented short review authors focused on recent findings on the immunological functions of intestinal mesenchymal cells. Authors сonsider significant problem, that close to gut's inflammatory diseases and regeneration. The figures are well-drawn. Unfortunately, the additional value of materials is not sufficient for a review paper. I believe the cause is that the authors have chosen too broad of a scientific field.
I recommend authors shift the focus to specific aspects, for example to chronic inflammatory bowel disease or infections.
Author Response
We thank the reviewer for this comment. While an average review includes 1-3 figures related to the discussed topic, in our review, we included 2 complex figures and 3 additional tables. These tables help the readers to orientate regarding the different subsets of intestinal mesenchymal stromal cells and endothelial cells and highlight one of the key messages of our review, namely the capacity of intestinal mesenchymal stromal cells in secreting a vast array of immunomodulatory factors whereby can actively form tissue niches for hematopoietic cells. This may occur both at homeostatic conditions and under inflammatory conditions, albeit with different consequences which is why we tried to include both scenarios. After all, it remained unclear to us, what exactly the above comment means: “the additional value of materials is not sufficient for a review paper”.
With regards to the last sentence in the reviewer`s comment suggesting a shift in the focus of the review, we cannot agree due to the following reasons:
- We have already discussed all current knowledge on the role of intestinal mesenchymal stromal cells in inflammatory bowel disease (paragraph 4.2) and infections (first part of paragraph 4.1) in our review.
- Due to a limited number of available publications, shifting the focus on only one or the other mentioned topic would not suffice to write a review paper. Especially the contribution of intestinal mesenchymal stromal cells to infections is hardly investigated, and beside from three in vivo studies (Ref #45: Kim et al., Immunity, 2011; Ref #65: Mahapatro et al., Cell Rep, 2016; Ref #80: Gibson et al., Gastroenterol, 2010) only in vitro data are available suggesting the responsiveness of intestinal mesenchymal stromal cells on PRR ligands.
- The field of intestinal mesenchymal stromal cells is a newly emerging and rapidly developing field, however, the available data in the literature would not allow focusing on the contribution of these cells to the pathogenesis of one particular inflammatory disorder. Besides, our review aims to demonstrate that these cells possess immunomodulatory properties in a wide variety of inflammatory conditions of the gastrointestinal tract.
Reviewer 2 Report
I read the manuscript entitled "Intestinal Mesenchymal – Immune Cell Niches in Mucosal Homeostasis and Immunity" with great interest. Actually, there are a large number of papers devoted to mesenchymal stromal cells, their differentiation and immunomodulatory properties. However, they are usually focused on mesenchymal cells that are easy to isolate and use for cell therapy purposes. A detailed analysis of the mesenchymal cell crosstalk in their natural niche is rather rare. To a large extent, this concerns the intestine, which has unique immunological features. So, I am absolutely sure that the topic will attract the readers’ attention. The review article is a thoroughly and well executed study which includes a lot of useful information. The text is well written and contains excellent illustrations. The English is good enough to understand the text clearly. The number of self-citations is minimal. The only small remark I have to make concerns Table 1.
It’s known that despite their origin from the embryonic mesenchyme, endothelial cells in adults are not classified as mesenchymal. The authors mentioned that in the introduction section (line 39). However, in Table1 titled "Intestinal mesenchymal stromal cell populations", two right columns are addressed to endothelial cells. In my opinion, the name of the table should be corrected to avoid the contradiction. Also, there are too many blank cells in the table. This makes the table look incomplete. It would be nice to fill them in, or maybe make them in a slightly different color to look “unfillable”. Considering these remarks, I believe that the manuscript can be published as it is.
Author Response
We thank the reviewer for this very important comment and apologize for the contradiction in the definition of mesenchymal stromal cells and the title of Table 1. We have corrected the name of Table 1 accordingly. Also, we have reduced the width of each column and filled the empty cells with light grey color to make them look less unfilled (this can only be seen in non-track mode).
Reviewer 3 Report
In this manuscript, the authors present a review of the literature on the role of intestinal mesenchymal cells on gut homeostasis and inflammatory and infectious conditions.I find that this review is generally of a good level. However, I would make a few comments.
I find the title is confusing by the use of mesenchymal – immune cell term.
The authors should introduce the fact that fibroblasts and myofibroblasts are the main focus of the review (maybe because of a lack of literature on the other IMC).
To my opinion, the authors should insist more on the characterization of IMCs by their spatial heterogeneity or by their surface markers. As an example, one might wonder the difference between trophocytes and Pdgfra lo stromal cells if not their location?
Line 32, instead of those 6 references, all dealing with focused immune populations, I would have chosen one or 2 reviews dealing with more general aspects of the gut immune response and tolerance.
Line 80, a recent reference on telocytes and wnt and shh signalling published in nature in 2022 is missing.
Line 81, the abbreviation used in the table for vascular endothelial cells is Va-EC.
Lines 85 and 87, I’m not sure about the reference number 19, in this context.
Some markers described in the text, such as Hhip, gp-38 or Ackr4 are not included in the table, is there a reason?
The table 1 presents IMC markers, it should not include endothelial cells. The Vim marker present in table 1 was not defined in the text.
Table 1 change Pdgfralo stroma by Pdgfra lo stromal cells.
Line 126: The authors indicate a location for the trophocytes “in subcryptal”, however the figure 1A is misleading and shows these cells rather at the level of the muscle layer.
These trophocytes are also responsible for a large part of the secretion of BMPs and yet they do not appear in figure 1B showing only telocytes (grey cells).
What do the question marks in Figure 2c mean?
Line 160 : The author should mention that IMC also contribute to education of mast cells in this paragraph.
Lines 216 and throughout the manuscript: The term perturbations should be avoided and better specified, for example by inflammatory and infectious conditions.
Line 223: the introduction of the role of IMC during infection is missing.
Line 242: Some words about the phagocytic and Antigen presenting cells capacities of IMC are missing.
Line 313: the term cells is missing.
Line 331 : the word disease is already included in IBD.
Line 380 : I do not agree with the term loss-of-function concerning NOD2 mutations.
Line 508: the reference is missing.
Some research articles and notions are, to my opinion, missing in this review. As exemples, recently Ding and colleagues, in Cell Research, showed the impact of Lepr+ mesenchymal cells surrounding intestinal crypts on gut homeostasis. Or the role of IMC on cell plasticity described by Melissari et al in Cellular and Molecular Life Sciences.
Author Response
In this manuscript, the authors present a review of the literature on the role of intestinal mesenchymal cells on gut homeostasis and inflammatory and infectious conditions.
I find that this review is generally of a good level. However, I would make a few comments.
I find the title is confusing by the use of mesenchymal – immune cell term.
We agree that the title was somewhat confusing, therefore we have rephrased the title and removed the “mesenchymal – immune cell” term.
The authors should introduce the fact that fibroblasts and myofibroblasts are the main focus of the review (maybe because of a lack of literature on the other IMC).
We thank the reviewer for this useful comment and added this information to the end of the “Introduction” (Lines 55-58).
To my opinion, the authors should insist more on the characterization of IMCs by their spatial heterogeneity or by their surface markers. As an example, one might wonder the difference between trophocytes and Pdgfra lo stromal cells if not their location?
Indeed, the characterization of IMC populations can be done either by their localization or by surface marker expression. In the paragraph about IMC heterogeneity, we added both information to provide the best characterization of each subset. This holds for trophocytes, which subset can be distinguished from Pdgfra low stromal cells due to their localization, but also via their high expression of certain surface markers (Grem1, CD81) makes them a separate population. We admit that the description of this population might have been confusing, therefore we have rephrased this part of the text (between Lines 94-98).
Line 32, instead of those 6 references, all dealing with focused immune populations, I would have chosen one or 2 reviews dealing with more general aspects of the gut immune response and tolerance.
We thank the reviewer for this suggestion and removed 5 out of 6 references from Line 32, kept 1 of these references which is a review on gut immune tolerance, and added a new review discussing general aspects of gut immunity. The original references #4 and #6 (Pezoldt et al., Nat Comm, 2018; Pasztoi et al., Eur J Immunol, 2017) dealing with the maintenance of intestinal tolerance by mLN stromal cells have been moved to Lines 220-221 (now Reference #66 and #67).
Line 80, a recent reference on telocytes and wnt and shh signalling published in nature in 2022 is missing.
We have added the mentioned reference (Maimets et al., Nat Comm, 2022) to Line 87 and also to Lines 220-223 as reference #13.
Line 81, the abbreviation used in the table for vascular endothelial cells is Va-EC.
We apologize for the inconsistency. We have corrected the abbreviation accordingly.
Lines 85 and 87, I’m not sure about the reference number 19, in this context.
We apologize for the incorrect insertion of this reference. We have anyways rephrased this part of the paragraph (see our answer to the next comment) thus the reference is now correctly moved to the previous sentence (Lines 91-95, current reference number #16).
Some markers described in the text, such as Hhip, gp-38 or Ackr4 are not included in the table, is there a reason?
For the sake of clarity, we have now added gp-38 (Pdpn) and Ackr4 to Table 1. We have not added Hhip to Table1, as the information about its expression in other subsets is unclear, thus rather removed this marker from the text and merged the two sentences about myocytes into one (Lines 86-90).
The table 1 presents IMC markers, it should not include endothelial cells. The Vim marker present in table 1 was not defined in the text.
We apologize for the misleading title of Table 1. We included endothelial cells in the title and defined Vim in the text (Lines 154-157).
Table 1 change Pdgfralo stroma by Pdgfra lo stromal cells.
We have changed the name of Pdgfralo stromal cells to “Pdgfra low stroma “in Table 1.
Line 126: The authors indicate a location for the trophocytes “in subcryptal”, however, the figure 1A is misleading and shows these cells rather at the level of the muscle layer.
Trophocytes are located beneath the crypt bottom along the external muscle layer as described in Lines 98-100. In vivo, however, crypts are densely located next to each other, which cannot be shown in the figure (where is a huge space next to the crypt to be able to show details within the villi), otherwise, further components of the figure would be lost and the whole figure would be too complicated. To make the “subcryptal” location look better, we have removed some of the space between the crypt and the muscle layers. We hope that this solution makes the localization of these cells clearer.
These trophocytes are also responsible for a large part of the secretion of BMPs and yet they do not appear in figure 1B showing only telocytes (grey cells).
Indeed, trophocytes are also key in producing for example Grem1 and shape the Wnt/BMP gradient along the crypt-villus axis. We now added these cells to Figure 1B.
What do the question marks in Figure 2c mean?
The reviewer probably meant Figure 1C, right? Originally, we used the question marks to express that the functions of IMC-derived cytokines and chemokines are not fully understood. For the sake of clarity, we have removed them.
Line 160: The author should mention that IMC also contribute to education of mast cells in this paragraph.
We thank the reviewer for this essential comment. We included a publication (Kurashima et al., Immunity, 2014) about the crosstalk between IMCs and mast cells in the mentioned paragraph (Lines 274-281, Ref #76).
Lines 216 and throughout the manuscript: The term perturbations should be avoided and better specified, for example by inflammatory and infectious conditions.
The term „perturbation” has been replaced throughout the manuscript by the terms “inflammation” or “inflammatory conditions”.
Line 223: the introduction of the role of IMC during infection is missing.
We added the missing information, which is now in Line 304.
Line 242: Some words about the phagocytic and Antigen presenting cells capacities of IMC are missing.
We added here 3 new publications (Zawahir et al., J Interferon Cyt Res, 2015; Saada et al., J Immunol, 2006; Koyama et al., Nat Med, 2011, References #59, #89, #90) indicating the antigen-presenting capacity of IMCs under steady-state and inflammatory conditions/cytokine stimulation (Lines 388-391).
Line 313: the term cells is missing.
We thank the reviewer for highlighting this typo. We added the missing word (Line 400).
Line 331 : the word disease is already included in IBD.
We removed the extra word „disease“ from the text (Line 418).
Line 380 : I do not agree with the term loss-of-function concerning NOD2 mutations.
We removed the term „loss-of-function” and now only mention NOD2 mutations throughout the paragraph.
Line 508: the reference is missing.
The sentence in Line 508 (now Line 620) included our observation during in vitro culture experiments, which we now made clear.
Some research articles and notions are, to my opinion, missing in this review. As exemples, recently Ding and colleagues, in Cell Research, showed the impact of Lepr+ mesenchymal cells surrounding intestinal crypts on gut homeostasis. Or the role of IMC on cell plasticity described by Melissari et al in Cellular and Molecular Life Sciences.
We thank the reviewer for the suggestion to include the above-mentioned recently published papers. The study by Deng et al. is now added and can be found in Lines 197-201 (Ref #37). The paper from Melissary et al. is now cited in Lines 496-507 (Ref #97).
Round 2
Reviewer 1 Report
Despite some additions to the publication, the revisions made are not enough to accept the paper. I guess the purpose of the study does not meet the requirements of high scientific quality and significance.